# The Antiviral Effect of Berdazimer Sodium on Molluscum Contagiosum Virus Using a Novel In Vitro Methodology

**DOI:** 10.3390/v15122360

**Published:** 2023-11-30

**Authors:** Brian M. Ward, Daniel A. Riccio, Martina Cartwright, Tomoko Maeda-Chubachi

**Affiliations:** 1Department of Microbiology and Immunology, University of Rochester Medical Center, Rochester, NY 14642, USA; brian_ward@urmc.rochester.edu; 2Novan, Durham, NC 27703, USA; driccio@novan.com (D.A.R.); mcartwright@novan.com (M.C.)

**Keywords:** molluscum contagiosum, molluscum contagiosum virus, nitric oxide, berdazimer sodium, mechanism of action

## Abstract

Molluscum contagiosum (MC) is characterized by skin lesions containing the highly contagious molluscum contagiosum poxvirus (MCV). MCV primarily infects children, with one US Food and Drug Administration (FDA)-approved drug-device treatment in use but no approved medications. Assessing antivirals is hindered by the inability of MCV to replicate in vitro. Here, we use vaccinia virus as a surrogate to provide evidence of the anti-poxvirus properties of berdazimer sodium, a new chemical entity, and the active substance in berdazimer gel, 10.3%, a nitric oxide-releasing topical in phase 3 development for the treatment of MC. We show that berdazimer sodium reduced poxvirus replication and, through a novel methodology, demonstrate that cells infected with drug-treated MCV virions have reduced early gene expression. Specifically, this is accomplished by studying the nuclear factor kappa-light-chain-enhancer of activated B cell (NF-kB)-blocking protein MC160 as an example of an early gene. The results provide a plausible unique antiviral mechanism of action supporting increased MCV resolution observed in patients treated with berdazimer gel, 10.3% and describe a novel methodology that overcomes limitations in investigating MCV response in vitro to a potential new MC topical medication.

## 1. Introduction

Molluscum contagiosum (MC) is a highly contagious skin infection caused by the molluscum contagiosum virus (MCV) [1]. MCV affects ~6 million people in the US annually, with mostly children and immunocompromised adults affected [2]. All four MCV subtypes exclusively infect humans and replicate in the cytoplasm of keratinocytes. The most common subtype, MCV Type I, encodes for 182 proteins with 105 being common to other orthopoxviruses [3]. Infection is characterized by clusters of skin- to red-colored papules with a central, umbilicated viral core; up to 100 lesions may be present in severe cases [4]. MCV infections replicate in the cytoplasm of—and are limited to—the epidermis, and keratinocyte proliferation manifests microscopically as hyalinized, aggregated molluscum bodies (Henderson–Paterson bodies) [1]. Keratinocytes differentiate from the stratum basale, and MCV virions can be found in this layer. As keratinocytes differentiate and migrate from the stratum basale to the stratum corneum layer, the virus is released at the skin surface where epidermal growth factor receptors are upregulated, fostering cell division [1]. Lesions commonly spread to other parts of the body via a scratching–autoinoculation cycle, and infections may persist for months to years, remaining contagious until lesions clear on their own or are removed by physical or chemical destruction methods administered by experienced healthcare providers [5]. Resolution relies on the activation of host immunity; however, MCV, surrounded by keratinocytes, may evade immune surveillance through this physical barrier and/or expression of specific immunomodulation proteins [6,7]. Disrupting the physical barrier through the repeated application of chemicals or by physical means is thought to trigger an immune response; the recently approved cantharidin drug-device combination product applied in the healthcare provider’s office is an example of one such chemical treatment [8]. However, this product, and similar methods, harbor concerning side effects [5,8]. Currently, US Food and Drug Administration-approved prescription medications indicated for the self-treatment of MCV remain elusive. Given the localized nature of the virus, a topical antiviral preparation would be more advantageous than a systemic treatment.

Developing new MCV therapeutic strategies is challenging due to the lack of suitable animal and tissue culture models in which to study the virus’ full replicative cycle. In tissue culture, MCV has been shown to enter cells and expresses early genes but does not undergo genome replication and the subsequent steps of virion assembly and release [9] Much of our knowledge of the MCV replicative cycle is inferred from studies on the related poxvirus vaccinia [10]. Furthermore, vaccinia virus has served as a surrogate for the study of MCV proteins, including the immune evasion proteins MC159 and MC160. MC159 and MC160 are homologs of each other and are expressed early during infection [11,12]. Thus, the MC160 protein has been identified as a potential MCV therapeutic target.

MC160 prevents the degradation of IκBα, an inhibitor of the nuclear factor kappa-light-chain-enhancer of activated B cells (NF-κB), thereby inhibiting the NF-κB pathway during early MCV infection [11,12,13]. NF-κB is a pleiotropic transcription factor residing in the cytoplasm, that, upon activation and translocation to the nucleus, regulates the transcription of genes involved in cell cycle, apoptosis, and cytokine production. MC160 has been shown to reduce tumor necrosis factor alpha (TNF-α)-mediated NF-κB activation involving tumor necrosis factor receptor-associated factor 2 (TRAF2), NF-κB-inducing kinase (NIK), and the myeloid differentiation primary response 88 (MyD88) adaptor protein.

MC160 may also inhibit interferon regulatory factor-3, a transcription factor necessary to produce antiviral interferon-β, a modulator of pro- and anti-inflammatory pathways. Decreased interferon-β levels may contribute to reduced immune response to MCV [7]. MC160 belongs to the viral Fas-associated death domain-like interleukin (IL)1-beta-converting enzyme (FLICE)-inhibitory protein (vFLIP), thus interfering with apoptotic pathways, in part by inhibiting procaspase activation [14].

Berdazimer gel, 10.3%, in late-stage clinical development as a potential first prescription medication specifically indicated for MC, contains berdazimer sodium, a new chemical entity. Berdazimer sodium is a polymeric drug substance consisting of a polysiloxane backbone (Si-O-Si bonds) with covalently bound *N*-diazeniumdiolate nitric oxide (NO) donors throughout (Figure 1). Exposure to a proton donor, such as water, promotes NO release from the polymeric drug substance via decomposition of the *N*-diazeniumdiolates [15] (Figure 1). Novan’s proprietary platform of NO-releasing technology, NITRICIL™, affords the stable storage of large amounts of NO onto the polymeric drug substance scaffold and circumvents the challenges associated with conventional low molecular weight NO donors, such as an inability to be formulated into a stable drug product [16]. In a randomized, controlled clinical trial, berdazimer gel, 10.3% resulted in significant reduction in molluscum lesion counts versus vehicle after 12 weeks of a once-daily application [17]. Subgroup analyses from the integrated data of phase 3 studies further confirmed the effect of berdazimer gel on MC lesion reduction in various subgroups, including age, sex, baseline lesion count, and disease duration [18]. Nitric oxide is known to have broad-spectrum antimicrobial and antiviral activity [19], and NO release from berdazimer sodium has been shown to have an antiviral effect against another virus, human papillomavirus [20]. However, the direct mechanism of action of berdazimer gel, 10.3% against MCV is unknown. The purpose of this study was to evaluate the potential mechanism of action of berdazimer sodium against poxviruses and to assess the effects of berdazimer sodium on the expression of molluscum immune evasion protein MC160. 

MCV is a dermatotropic poxvirus that has a virion structure and genome similar to the prototypical poxvirus vaccinia virus [3,7,12,21,22]. MCV does not replicate in tissue culture but encodes numerous homologs to vaccinia virus and, therefore, is predicted to replicate in a manner similar to vaccinia virus. Consequently, assaying virus replication by measuring progeny virion production is not possible for MCV. Therefore, we first evaluated the antiviral properties of berdazimer sodium on vaccinia virus. After evaluating the cytotoxicity of berdazimer sodium, two different experiments were conducted to determine the effect of the drug on vaccinia virus replication. The first study was to infect cells treated with a non-toxic dose of berdazimer sodium with a low multiplicity of infection of vaccinia virus and titer the resulting progeny virions. The second study was to treat purified virions directly with the drug prior to infection to determine if the drug had a direct effect on the infectivity of the virus particle. Although MCV does not replicate in vitro, it does proceed through earlier stages (cell entry and early gene expression) of the virus life cycle in tissue culture cells. A known early gene expressed by MCV is MC160, which is known to aid MCV in modulating and evading the host immune response [12]. Therefore, the expression of the MCV early gene MC160 was measured by Western blot to determine if there was an antiviral effect on MCV after a treatment of MCV virions with berdazimer sodium.

## 2. Materials and Methods

*Compounds.* Berdazimer sodium (CAS 1846565-00-1) was supplied by Novan and stored at −20 °C. Mitomycin C was purchased from Fisher Scientific (Waltham, MA, USA). Tissue culture grade DMSO was purchased from Corning Scientific (Corning, NY, USA). CellTiter-Glo was obtained from Promega (Madison, WI, USA).

*Cell culture.* The BSC40 cell line (African green monkey, kidney) was obtained from ATCC (Manassas, VA, USA) (CRL-2761) and maintained in DMEM supplemented with 8% Cosmic Calf Serum (Hyclone, Logan, UT, USA).

*Virus.* A recombinant reporter Western Reserve strain of vaccinia virus (VVEGIR), which expresses a green fluorescent protein (mNeonGreen) under an early promotor, and a red fluorescent protein (mKate2) under an intermediate promoter were used for all experiments. Curetted MC lesions were obtained with consent from a dermatologist. MCV particles were extracted from lesions using a bead mill and purified over a 36% sucrose cushion. Purified particles were resuspended in 0.1 M Tris. Genomic DNA was extracted using the Wizard MiniPrep kit (Promega) and quantified via quantitative polymerase chain reaction using oligos and probes directed against the viral polymerase gene MC039L.

*Drug toxicity.* A count of 1 × 10^4^ BSC40 cells were seeded into each well of a 96-well plate and incubated overnight at 37 °C. The following day, cell media was removed and replaced with 100 μL of fresh DMEM with 20 mM 2-[4-(2-Hydroxyethyl)piperazin-1-yl]ethane-1-sulfonic acid (HEPES), adjusted to pH 6.5 (DMEM6.5). Berdazimer sodium was mixed fresh by resuspending 100 mg in 500 μL of DMSO to provide a starting concentration of 200,000 μg/mL. This was diluted to 10,000 μg/mL in DMEM with 20 mM HEPES and adjusted to pH 4.0 (DMEM4). Note, the berdazimer sodium drug substance has an inherent buffering capacity that can shift the pH to alkaline upon reconstitution in solutions. The shift in pH needs to be accounted for upon suspension and subsequent dilution to ensure cell viability and release of NO from the *N*-diazeniumdiolate groups of berdazimer sodium. The use of DMEM4 in these experiments was to yield a final, target pH of 6.5 upon the resuspension of berdazimer sodium at these specified concentrations. This was then diluted to 2× the starting concentration for serial dilution in DMEM6.5. Equal volumes of the starting concentrations were added to the first row of cells in a 96-well plate and 1:2 serial dilutions were made down the plate. DMSO and mitomycin C were added at a 7/1000 dilution and 125 μg/mL to control wells. Treated cells were incubated overnight at 37 °C. The next day, the cells were washed 1× in phosphate-buffered saline. A total of 100 μL of fresh DMEM was added, followed by 100 μL of CellTiterGlo. This was shaken for 2 min and then transferred to an opaque 96-well plate. Luminescence was read using a luminometer with a 1 s integration time. Data were exported to MS Excel and analyzed.

*Low multiplicity-of-infection growth curve.* A count of 5 × 10^5^ BSC40 cells were seeded into each well of a 12-well plate and incubated overnight at 37 °C. The following day, berdazimer sodium was mixed fresh by resuspending 100 mg in 500 μL of DMSO to give a starting concentration of 200,000 μg/mL. This was diluted to 10,000 μg/mL in DMEM4. The use of DMEM4 in these experiments was to yield a final, target pH of 6.5 upon resuspension of the alkaline berdazimer sodium at these specified concentrations. This was then diluted to 200, 100, 50, and 25 μg/mL in DMEM6.5. Media on the cells were removed and replaced with media containing the diluted drug. In one set of wells, media were replaced with DMEM6.5, containing 6 μL of DMSO/mL of media. Treated cells were incubated at 37 °C for 30 min. After 30 min, 1 × 10^3^ plaque-forming units of VVEGIR were added to each well, and the cells were returned to 37 °C. The following day, the wells were dosed again with berdazimer sodium as above. Three days post infection, the cells in each well were harvested by scraping and transferred to a 1.5 mL screwcap tube. The virus in the cells was released by 3 freeze–thaw cycles and titered by serial dilution and plaque assay on monolayers of BSC40 cells. Three days post infection, the monolayers were stained with crystal violet, and the number of plaques in the well was counted. This assay was repeated twice with 3 replicates each time. Data were entered into Microsoft Excel for analysis.

*Direct particle treatment.* Berdazimer sodium was mixed fresh by resuspending 100 mg in 500 μL of DMSO to give a starting concentration of 200,000 μg/mL. This was diluted to 1000 μg/mL in DMEM4. The use of DMEM4 in these experiments was to yield a final, target pH of 6.5 upon the resuspension of alkaline berdazimer sodium at these specified concentrations. This was then diluted to 400, 200, 100, and 50 μg/mL in DMEM6.5. A total of 100 μL containing 5 × 10^4^ plaque-forming units of vaccinia virus (VVEGIR) in DMEM6.5 was added to a tube, followed by 100 μL of the dilutions of berdazimer sodium above. The mixture was incubated overnight at room temperature. The next day, the treated virions were serially diluted 1/10 in pH 7.4 DMEM (DMEM7.4) four times (1/10,000), and the number of infectious particles in the dilutions was determined via plaque assay on monolayers of the BSC40 cells. Two days post infection, the monolayers were stained with crystal violet, and plaques were quantified. This assay was repeated twice with 3 replicates each time. Data were entered into Microsoft Excel for analysis. Note, DMEM7.4 was used in these experiments to optimize the infection of BSC40 cells with drug-treated vaccinia virus particles. The berdazimer sodium drug was diluted out at this point in the experiment, and a target pH of 6.5 was no longer necessary.

*Western blot.* Purified MCV, which contained 5 × 10^5^ genome copies, was suspended in 500 μL of DMEM6.5. The suspension of purified MCV in DMEM6.5 was aliquoted equally between 5 tubes. Berdazimer sodium was mixed fresh in DMSO at a starting concentration of 200,000 μg/mL. This was diluted to 10,000 μg/mL in DMEM4. The use of DMEM4 in these experiments was to yield a final, target pH of 6.5 upon the resuspension of alkaline berdazimer sodium at these specified concentrations. This was then diluted to 2× the final concentration in DMEM6.5, and an equal volume (i.e., 100 μL) was added to the aliquoted virions to give final concentrations of 50, 100, 200, and 400 μg/mL. Treated virions were incubated at room temperature overnight. The next day, 800 μL of DMEM, adjusted to pH 7.4 (DMEM7.4), was added to the treated virions, and the entire volume was inoculated onto 1 × 10^5^ BSC40 cells. The extent of dilution was not as significant as performed for the drug-treated vaccinia virions to increase sensitivity for the ultimate Western blot readout. Inoculated cells were incubated at 37 °C overnight. The next day, the cells were harvested by scraping, washed in phosphate-buffered saline, and lysed in radioimmunoprecipitation buffer. Lysed cells were incubated 20 min on ice before being clarified by centrifugation at 18,000× *g* for 20 min. Clarified lysates were analyzed by Western blot, using anti-MC160 polyclonal anti-sera followed by horseradish peroxidase-conjugated donkey anti-rabbit antibody and mouse anti-actin, followed by Alexa-647-conjugated donkey anti-mouse antibody. Horseradish peroxidase and fluorescence signals were detected using a Kodak Image Station 40000MM Pro (Carestream Health Inc. (Rochester, NY, USA)). DMEM7.4 experiments were used in these experiments to optimize the infection of BSC40 cells using drug-treated MCV virus particles. The berdazimer sodium drug was diluted out at this point in the experiment, and a target pH of 6.5 was no longer necessary. Anti-MC160 polyclonal anti-sera was a kind gift from Joanna Shisler [23].

*Data analysis.* Band intensities were determined using imageJ (Bethesda, MD, USA). MC160 expression was normalized by dividing the calculated signal of the MC160 band in each lane with the corresponding actin signal in the same lane. Percent reduction was then calculated by dividing the normalized MC160 signal in each lane to the normalized MC160 signal in the 0 μg/mL (DMSO control) lane.

## 3. Results

### 3.1. Drug Toxicity

The treatment of BSC40 cells with varying concentrations of berdazimer sodium showed a dose response for toxicity (Figure 2). Plotting out the relationship between concentrations of drug and percent toxicity compared to DMSO treatment resulted in responses with an R^2^ value of 0.9169. Using linear regression, the concentration at which 50% of cells show cytotoxicity (CC_50_) was calculated to be 396.4 µg/mL.

### 3.2. Low Multiplicity-of-Infection Growth Curve

The infection of BSC40 cells that had been treated with varying concentrations of berdazimer sodium reduced vaccinia virus replication in a dose-dependent response as measured by virion production (Figure 3). Plotting the relationship between drug concentrations and the percent of the remaining titer compared to DMSO treatment resulted in a response with an R^2^ value of 0.8441. Using linear regression, the drug concentration that inhibited 50% of vaccinia virus production in cell culture (IC_50_) was calculated to be 37.43 μg/mL.

### 3.3. Direct Particle Treatment

The treatment of vaccinia virus virions with varying concentrations of berdazimer sodium prior to infection showed a dose-dependent reduction in virus infectivity as measured via plaque assay (Figure 4). Plotting the relationship between drug concentrations and percent titer compared to DMSO treatment resulted in a response with an R^2^ value of 0.9694. It was calculated that 404.6 μg/mL of the drug would be required to achieve the vaccinia virus IC_50_.

### 3.4. Western Blot

MCV particles treated with 50–400 μg/mL of berdazimer sodium showed decreased levels of the protein MC160, which is expressed by the MC160 early gene of MCV, as measured using Western blotting (Figure 5). The masses in kilodaltons and positions of marker proteins are shown on the sides of the blot. The analysis of signal intensity and normalization to the amount of actin to account for differences in lane loading verifies a dose-dependent response.

A response between drug concentration and percent of the actin-normalized MC160 expression compared to DMSO treatment was observed, with an R^2^ value of 0.9671 (Figure 6). Using linear regression and the equation calculated by MS Excel, the drug concentration required to achieve the IC_50_ for MC160 gene expression was calculated to be 192.9 μg/mL.

## 4. Discussion

Berdazimer sodium is a new chemical entity in phase 3 clinical development as a topical gel for the treatment of MC. A randomized, controlled trial in 891 patients with MC revealed berdazimer gel, 10.3% significantly reduced MC lesion count compared with a vehicle control [17]. Specifically, 32.4% of the berdazimer group achieved a complete clearance of MC lesions at week 12 compared with 19.7% of the vehicle group. Berdazimer gel was well tolerated. The most common adverse events were application-site pain and erythema, and adverse events led to discontinuation in 4.1% of the berdazimer group and 0.7% of the vehicle group. Despite the demonstrated clinical efficacy of berdazimer gel, 10.3%, the mechanism of action of berdazimer is unknown [19]. The experiments described herein used a novel methodology to elucidate possible mechanisms of action to account for the clinical findings.

MC is challenging to study under standard in vitro antiviral assay conditions (i.e., the measurement of reduction in progeny virus) due to its inability to propagate in vitro. Thus, vaccinia virus serves as a surrogate for studies involving poxviruses including MCV [7,24,25,26]. To further our understanding of the effect of berdazimer sodium on MCV, we developed and employed a novel methodology to examine the antiviral effect of berdazimer sodium, specifically through its impact on the expression of the MC160 protein.

By directly treating virion particles with berdazimer sodium, we were able to specifically evaluate viral targets in the absence of confounding cellular targets and demonstrated unequivocal antiviral properties of berdazimer sodium on both vaccinia virus and MCV. Our approach to directly evaluating the impact of berdazimer sodium against purified MC virions allowed for unique insights regarding the way MCV behaves in response to a specific therapeutic agent, which has been historically challenging to investigate both in vitro and in vivo. 

Berdazimer sodium releases NO in a pH-dependent manner when initiated by a proton. In these experiments, the pH of the media was designed to be pH 6.5 to initiate NO release without adversely affecting the cells or virus (i.e., contribute to toxicity). Cell toxicity and direct antiviral effects were distinguished by first establishing cytotoxicity levels in BSC40 cells, which resulted in an average 50% cytotoxicity concentration of ~400 μg/mL. Using this as a guide, the antiviral effect of the drug on the complete viral replication cycle was analyzed by measuring the amount of vaccinia virus produced after 3 days of berdazimer sodium exposure. In two separate experiments, berdazimer sodium concentrations of 50 μg/mL reduced the amount of vaccinia virus production by more than 50% (the IC_50_ was calculated to be 37.43 μg/mL). This effective concentration is well below the calculated 50% cytotoxicity concentration and demonstrated the ability of the drug to repress poxvirus replication. 

Next, viral and cellular drug targets were differentiated. To examine the ability of berdazimer sodium to directly target the virus, vaccinia virus virions were treated with varying concentrations of the drug. Treated samples were diluted 10,000-fold, and the remaining infectivity was determined using a plaque reduction assay and compared to virions treated with only DMSO as a control. These two independent assays showed that concentrations of ~400 μg/mL of berdazimer sodium could inactivate ~50% of the input virions. We theorized that the reduced virion infectivity was not due to residual drug cytotoxicity, as the treated virus was diluted 10,000-fold before being added to the cells. This reduced the drug concentrations to below 0.1 μg/mL, which is far below our established 50% cytotoxicity concentration for berdazimer sodium against BSC40 cells. These experiments demonstrated that berdazimer sodium directly acts on poxvirus particles to inactive them and inhibit infection. Specifically, as the virion was the only target when incubated with berdazimer sodium in these assays, any downstream impact on virus infectivity or progeny production must have occurred through a direct antiviral effect on the virion from berdazimer sodium. 

Finally, the direct effect of berdazimer sodium on MCV was investigated using a methodological approach that overcame the inability of MCV to replicate in vitro and the use of progeny viral titer as a readout for an antiviral effect. Specifically, adapting the same approach utilized with vaccinia, MCV virions were treated with berdazimer sodium, and early gene expression was used as a readout for antiviral effect. The reduced expression of MC160 observed was likely due to an overall reduction in the infectivity of the virions in response to treatment with berdazimer sodium, akin to what was seen in the vaccinia virus experiments. 

Indeed, the IC_50_ was calculated to be 192.9 μg/mL, and the diluted concentration of berdazimer sodium during incubation with the BSC40 host cells (i.e., after treatment of the MCV virions) was not higher than 80 μg/mL, thus mitigating any concern that the IC_50_ was confounded by cytotoxicity, and further supporting the theory that berdazimer sodium had a direct antiviral effect on MCV.

The MC160 protein allows MCV to modulate the TNF-α-induced activation of NF-κB, a key pathway of the host immune response [12]. A modulation of the antiviral immune response by MC160 likely lends to MCV persistence. Berdazimer sodium may elicit its clinical effect, in part, by the inhibition of MC160 gene expression and thus lead to the resolution of MCV infections via inflammation induced by the activation of NF-κB. Clinically, the inflammation of MC lesions, called the beginning-of-the-end sign, is recognized as a herald of MC resolution (i.e., clearance of lesion) [27]. Phase 3 studies conducted with berdazimer gel showed increased MC lesion clearance as the beginning-of-the-end sign intensified with treatment [28]. 

## 5. Conclusions

Molluscum contagiosum is a common skin infection primarily affecting children. Prescription medications indicated for the treatment of MC remain elusive, as in vitro and in vivo investigations of MCV are technically challenging. This study described a novel methodology to investigate the therapeutic effects of berdazimer sodium, a new chemical entity poised as a first-in-class topical prescription gel formulation, berdazimer gel, 10.3% specifically indicated for the treatment of MC. Berdazimer sodium, an NO-releasing agent, was shown to have antiviral activity through a reduced viral load of vaccinia, a poxvirus surrogate for MCV, and reduced the downstream expression of the immunomodulation protein MC160 without the induction of cytotoxicity. The results herein provide a plausible mechanistic explanation for the clinical results observed in the phase 3 study of berdazimer gel, 10.3%, [17] and suggest that one pathway by which berdazimer sodium may elicit its antiviral effect is through direct action on virions. Further studies are needed to confirm this finding; however, these experiments suggest the mechanism of action of berdazimer may differ from antiviral therapeutics that act on virion-associated enzymes or cellular targets [29].

Nitric oxide is known to have broad-spectrum antiviral activity [30] and, as such, the specific mechanism by which NO exerts its antiviral activity has been postulated to be through the *S*-nitrosylation of key cysteine residues within viral proteins. Future studies evaluating the specific site of action from berdazimer sodium on pox viral targets could further elucidate and support the mechanism of action for the antiviral activity described herein. Furthermore, the broad-spectrum antiviral activity associated with NO suggests that additional viral proteins and early genes, beyond that of MC160, may potentially be affected post berdazimer sodium-treatment. Future studies to establish the potential for additional viral targets of MCV are warranted.

In summary, this report highlights the use of a novel in vitro methodology to elucidate the antiviral effects of berdazimer sodium in the treatment of MC and contributes to a greater understanding of this complex and bothersome viral skin infection.

## Figures and Tables

**Figure 1 viruses-15-02360-f001:**
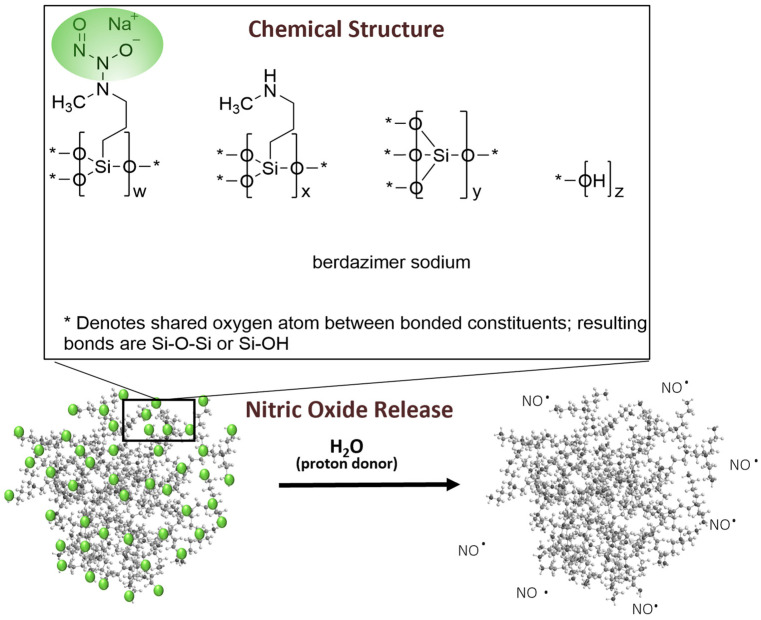
Berdazimer sodium. The chemical structure of berdazimer sodium comprises three distinct repeating silane units, one of which contains the NO-releasing *N*-diazeniumdiolate moiety. The polymeric berdazimer sodium drug substance releases NO upon exposure to a proton donor such as water.

**Figure 2 viruses-15-02360-f002:**
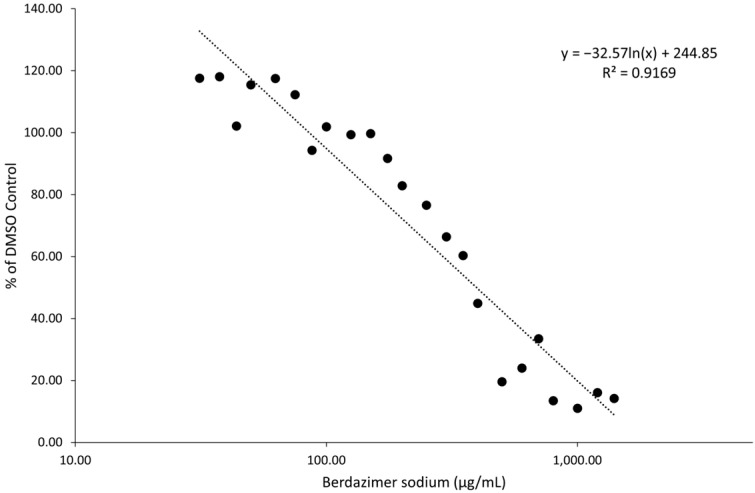
Cytotoxicity of berdazimer sodium to BSC40 cells. Data are presented as the average of *n* = 3 biological replicates, each comprising *n* = 3 technical replicates.

**Figure 3 viruses-15-02360-f003:**
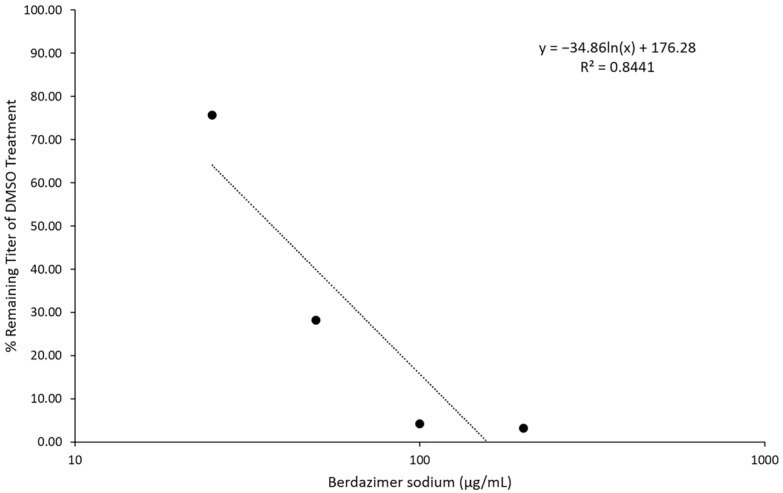
Effect of berdazimer sodium on vaccinia virus production. Inhibition of vaccinia virus production after exposure to berdazimer sodium. Data are presented as the average of *n* = 2 biological replicates, each with *n* = 3 technical replicates.

**Figure 4 viruses-15-02360-f004:**
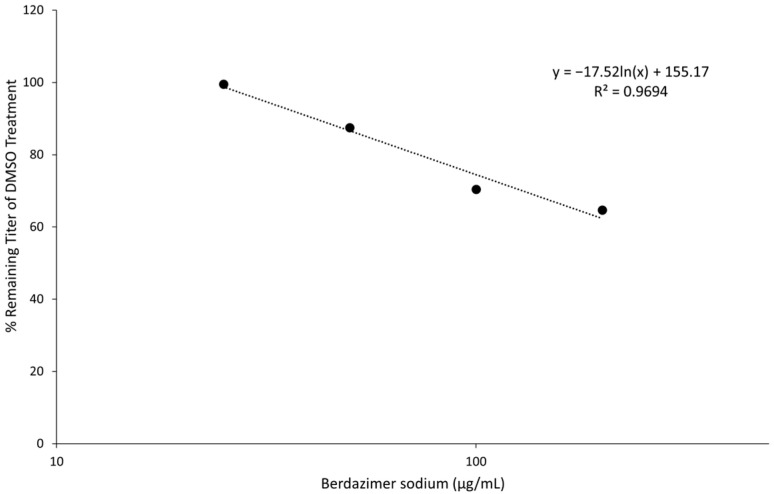
Effect of berdazimer sodium on vaccinia virus infectivity. Inhibition of vaccinia virus particle infectivity after an exposure of vaccinia virions to berdazimer sodium. Data are presented as the average of *n* = 2 biological replicates, each with *n* = 2–3 technical replicates.

**Figure 5 viruses-15-02360-f005:**
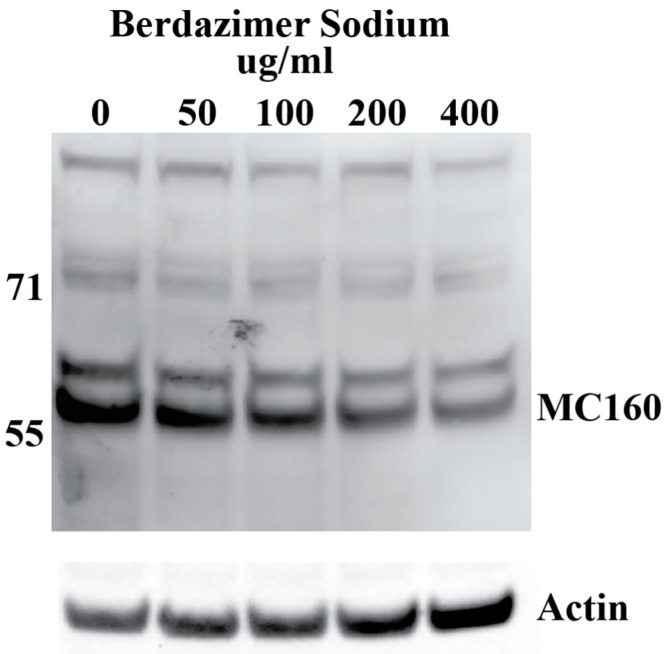
Western blot of MC160 gene expression after the exposure of molluscum contagiosum virions to berdazimer sodium.

**Figure 6 viruses-15-02360-f006:**
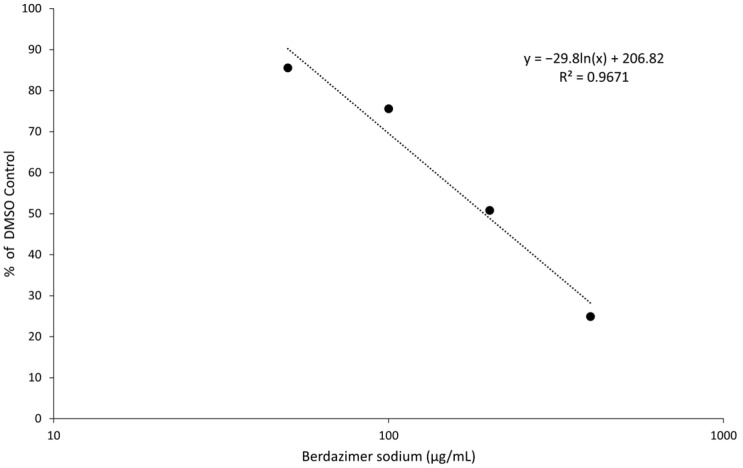
Inhibition of MC160 gene expression after the exposure of molluscum contagiosum virions to berdazimer sodium. Data are from one biological replicate.

## Data Availability

No datasets were generated or analyzed during this current study.

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
