# Peer review of "The Antiviral Effect of Berdazimer Sodium on Molluscum Contagiosum Virus Using a Novel In Vitro Methodology"

_viruses, 2023, doi:10.3390/v15122360_

Round 1

Reviewer 1 Report

Comments and Suggestions for Authors

This is a straight forward and well written study that investigates how berdazimer sodium might be Impacting poxvirus infection in cell culture.  As an initial surrogate for MCV infection, the researchers first study the compound’s effect on vaccinia virus infection.  They show that  the drug decreases productive infection at a concentration about 10-times lower than cellular toxicity.  Incubating vaccinia virions with a much higher concentration of the drug also decreases the amount of virus produced after infecting cells.  They then study the drug’s effect on MCV, a virus that does not replicate in cell culture.  They incubate the drug with MCV virions and show by western blot that the drug affects a step that leads to less early gene expression after an overnight infection with MCV through yet an unknown mechanism (decreased entry and/or a post-entry step).

Major points for the authors to consider:

Does berdazimer sodium also decrease vaccinia virus early gene expression similar to what was seen with pre-treatment of MCV virions?

Figure 5.  Western blot of cell lysates after overnight infection with MCV.  Conclusions about the effect on early gene expression could be strengthened by additional time points.  E.g., would data be same (better/worse?) if cells harvested 8 hours after infection?

Abstract, line 18.  I do not think the current work “establishes” the anti-poxvirus properties of berdazimer.  It provides evidence of anti-poxvirus activity.

Abstract, Line 22.  Since MC160 was used as a surrogate for MCV early gene expression, in the abstract I think it would be more correct to indicate the data shows a reduction in early gene expression.  Later in the paper the authors can septulate about what suppression of early gene expression might lead to (e.g., decreasing expression of a NF-kB blocking protein as well as many other proteins involved in a productive infection).  How th

Minor points to consider:

1. Methods or results. While ultimately explained in the discussion, authors should provide some insight onwhy various DMEMs were used (DMEM7.4, DMEM6.5, DMEM4)

 2. Line 198. Source of anti-MC160 polyclonal anti-sera. Reference or description on how it was generated.

 3. Line 262.  Would be helpful to contain a sentence or two about the results of the JAMA Dermatol publication For example, 32.4% of the berdazimer group achieved complete clearance of MC lesions at week 12 compared with 19.7% of vehicle treated controls.  Adverse events leading to discontinuation affected 4.1% of the berdazimer group and 0.7% of the vehicle group.  Also note a reference mishap in the manuscript - line 262:  Browning not Brown.

 4. Line 287. Earlier in the manuscript authors indicated IC50 to be 37.43 μg/mL. Why is text now saying 50 μg/mL?

 5. Line 314-318.  Why was the calculated dose of berdazimer sodium after direct treatment of vaccinia virions to be below 0.1 μg/mL and for MCV it  was so much higher (not higher than 80 μg/mL)?

Reviewer 2 Report

Comments and Suggestions for Authors

The authors provided a new approach to assess the antiviral effect of berdazimer sodium specifically through its impact on the expression of MC160 protein.

The manuscript is clear and the methodology well described. I suggest adding a recent reference from your group (DOI: 10.1016/j.jaad.2023.09.066). Are there any studies planned or underway that evaluate an extension of the treatment beyond 12 weeks? Based on your experiments, is it possible to hypothesize a change in dosage?

Round 2

Reviewer 1 Report

Comments and Suggestions for Authors

This is a revised manuscript where authors adjusted text in manuscript to address comments.  No additional experimental results were provided.

For this journal, authors have adequately responded to comments.

Upon re-reading the abstract, line 10 needs to be corrected. There is an FDA-approved treatment for Molluscum (https://www.fda.gov/drugs/news-events-human-drugs/fda-approves-first-treatment-molluscum-contagiosum) which the authors state in line 45. Would suggest deleting mention of FDA approval of drugs in the abstract.

Author Response

November 23, 2023

Editor-in-Chief, Viruses

Director, HIV Dynamics and Replication Program

Center for Cancer Research

National Cancer Institute

Frederick, MD 21702-1201

Dear Dr. Freed,

Thank you for the opportunity to finalize our article “Antiviral Effect of Berdazimer Sodium on Molluscum Contagiosum Virus Using a Novel In Vitro Methodology.” Please find our point-by-point responses to the final peer reviewer’s comments below, and we have uploaded a highlighted version of our manuscript that indicates these revisions.

As with our original submission, we confirm that neither the manuscript nor any parts of its content are currently under consideration or published in another journal. All authors have approved the revised manuscript and agree with its resubmission to Viruses.

Warm Regards,

Tomoko Maeda-Chubachi, MD, PhD, MBA

Chief Medical Officer

Novan

tmaeda-chubachi@novan.com

tomokomaeda@hotmail.com

Mobile: 858-539-6783

Reviewer's Comments

Comments and Suggestions for Authors
This is a revised manuscript where authors adjusted text in manuscript to address comments. No additional experimental results were provided.

For this journal, authors have adequately responded to comments.

Upon re-reading the abstract, line 10 needs to be corrected. There is an FDA-approved treatment for Molluscum (https://www.fda.gov/drugs/news-events-human-drugs/fda-approves-first-treatment-molluscum-contagiosum) which the authors state in line 45. Would suggest deleting mention of FDA approval of drugs in the abstract.

Response

Thank you for catching this inconsistency. We have addressed the reviewer’s comment in line 10 of the abstract to be consistent with that of line 45 (now lines 46-47) to read “with one US Food and Drug Administration (FDA)–approved drug-device treatment but no approved medications.”

The recently approved cantharidin drug-device is the first FDA-approved treatment procedure for molluscum; it is a treatment that requires in-office administration. The language in the amended version highlights the fact that there are no approved FDA medications. As described in the body text, if approved, berdazimer gel would be the first FDA approved prescription medication for molluscum. This is an important distinction between the two products, as one is a drug-device treatment and the other is a medication. We also added the word “treatment” to line 47.

Also, we corrected the spelling of Dr. Maeda-Chubachi’s name in the author line as highlighted.
